DnoisE: distance denoising by entropy. An open-source parallelizable alternative for denoising sequence datasets

Antich Adrià 1
Palacín Creu 2
Turon Xavier 1
Wangensteen Owen S. owen.wangensteen@uit.no 3
1 Department of Marine Ecology, Centre for Advanced Studies of Blanes (CEAB- CSIC) , Blanes (Girona) , Catalonia , Spain
2 Department of Evolutionary Biology, Ecology and Environmental Sciences and Biodiversity Research Institute (IRBIO), University of Barcelona , Barcelona , Catalonia , Spain
3 Norwegian School of Fishery Science, UiT The Arctic University of Norway , Tromsø , Troms og Finnmark , Norway
Gillespie Joseph
Electronic publication date: 2022 Jan 19
Publication date: 2022
Volume: 10
Electronic Location ID: e12758
Received 2021 Jul 13; Accepted 2021 Dec 16
Copyright: ©2022 Antich et al.
Copyright year: 2022
Copyright holder: Antich et al.
License: This is an open access article distributed under the terms of the Creative Commons Attribution License, which permits unrestricted use, distribution, reproduction and adaptation in any medium and for any purpose provided that it is properly attributed. For attribution, the original author(s), title, publication source (PeerJ) and either DOI or URL of the article must be cited.
License URL: https://creativecommons.org/licenses/by/4.0/

Keywords: Metabarcoding, Bioinformatic pipelines, Metaphylogeography, Entropy correction, Denoising algorithms, Coding markers

Funding: PopCOmics CTM2017-88080 MCIN/AEI/10.13039/501100011033 MARGECH PID2020-118550RB MCIN/AEI/10.13039/501100011033 BigPark from the Spanish Government OAPN, 2462/2017 UiT The Arctic University of Norway This research was funded by the projects PopCOmics (CTM2017-88080, MCIN/AEI/10.13039/ 501100011033 and “ERDF A way of making Europe”, EU), MARGECH (PID2020-118550RB, MCIN/AEI/10.13039/501100011033), and BigPark (OAPN, 2462/2017) from the Spanish Government. The publication charges for this article have been funded by a grant from the publication fund of UiT The Arctic University of Norway. The funders had no role in study design, data collection and analysis, decision to publish, or preparation of the manuscript.

==============================
DNA metabarcoding is broadly used in biodiversity studies encompassing a wide range of organisms. Erroneous amplicons, generated during amplification and sequencing procedures, constitute one of the major sources of concern for the interpretation of metabarcoding results. Several denoising programs have been implemented to detect and eliminate these errors. However, almost all denoising software currently available has been designed to process non-coding ribosomal sequences, most notably prokaryotic 16S rDNA. The growing number of metabarcoding studies using coding markers such as COI or RuBisCO demands a re-assessment and calibration of denoising algorithms. Here we present DnoisE, the first denoising program designed to detect erroneous reads and merge them with the correct ones using information from the natural variability (entropy) associated to each codon position in coding barcodes. We have developed an open-source software using a modified version of the UNOISE algorithm. DnoisE implements different merging procedures as options, and can incorporate codon entropy information either retrieved from the data or supplied by the user. In addition, the algorithm of DnoisE is parallelizable, greatly reducing runtimes on computer clusters. Our program also allows different input file formats, so it can be readily incorporated into existing metabarcoding pipelines.

Background

Biodiversity studies have experienced a revolution in the last decade with the application of high throughput sequencing (HTS) techniques. In particular, the use of metabarcoding in ecological studies has increased notably in recent years. For both prokaryotic and eukaryotic organisms, a large number of applications have been developed, ranging from biodiversity assessment (Wangensteen et al., 2018), detection of particular species (Kelly et al., 2014), analysis of impacts (Pawlowski et al., 2018), and diet studies (Clarke et al., 2020; Sousa, Silva & Xavier, 2019), among others. Also, different sample types have been used: terrestrial soil, freshwater, marine water, benthic samples, arthropod traps, or animal faeces (Creer et al., 2016; Deiner et al., 2017). Many of these studies have direct implications on management and conservation of ecosystems and are thus providing direct benefits to society. They have also brought to light a bewildering diversity of organisms in habitats difficult to study with traditional techniques.

Metabarcoding studies have greatly contributed to so-called big community data (Pichler & Hartig, 2020) by generating an enormous amount of sequence data that, in most cases, is available online. Handling these datasets is memory intensive and filtering steps are required to analyze such information. Clustering and denoising are the two main strategies to compress data into Molecular Operational Taxonomic Units (MOTUs, aka OTUs) or Exact Sequence Variants (ESVs; also ASVs, Amplicon Sequence Variants, or ZOTUs, zero ratio OTUs) to extract biodiversity composition (Antich et al., 2021). Both methods rely on minimizing sequencing and PCR errors either by clustering sequences into purportedly meaningful biological entities (MOTUs) or by merging erroneous sequences with the correct ones from which they possibly originated, and keeping just correct amplicons (ESVs). Hence, both methods differ philosophically and analytically. Furthermore, they are not incompatible and can be jointly applied. Software development is crucial to create tools capable of performing these tasks in a fast and efficient way. The type of samples, the marker, and the target organisms are also instrumental in choosing the adequate bioinformatic pipelines to provide interpretable results.

Recent studies have explored the joint application of both methods to filter metabarcoding data (Antich et al., 2021; Brandt et al., 2021; Elbrecht et al., 2018; Turon et al., 2020). Importantly, the combination of clustering and denoising opens the door to the analysis of intraspecies (intra-MOTU) variability (Antich et al., 2021). Turon et al. (2020) proposed the term metaphylogeography for the study of population genetics using metabarcoding data, and Zizka, Weiss & Leese (2020) found different haplotype composition between perturbed and unperturbed rivers, both studies using a combination of clustering and denoising steps.

The software presented here focuses on the denoising step. There are currently several software programs developed to denoise sequencing and PCR errors, such as DADA2 (Callahan et al., 2016), AmpliCL (Peng & Dorman, 2020), Deblur (Amir et al., 2017), or UNOISE (Edgar, 2016). These programs have been widely used in metabarcoding studies to generate ESVs, using sequence quality information for the first two and simple analytical methods for the latter two. All were originally tested for ribosomal DNA (non-coding) and thus some adjustment is necessary for application to other markers (Antich et al., 2021).

Here we present DnoisE, a parallelizable Python3 software for denoising sequences using a modification of the UNOISE algorithm and tested for metabarcoding of eukaryote communities using mitochondrial markers (COI, Cytochrome Oxidase subunit I). We introduce a novel correction procedure for coding sequences using changes in diversity values per codon position. In coding genes, the natural entropy of the different positions is markedly different, with the third position being always the most variable. We therefore contend that differences in each position should have different weights when deciding whether a change in a given position is legitimate or is attributable to random PCR or sequencing errors. DnoisE is also applicable to other markers due to the settable options and offers a fast and open source alternative to non-parallelizable closed source programs. Scripts for installation and example files to run DnoisE are provided in the GitHub repository: https://github.com/adriantich/DnoisE.

Workflow

Structure of input files

DnoisE is designed to run with HTS datasets (after paired-end merging and de-replicating sequences) to obtain ESVs, or after clustering with SWARM (Mahé et al., 2015) to obtain haplotypes within MOTUs. Due to variability in format files, we have designed an algorithm that can read both fasta and csv files. In the present version, however, sample information (if present) is kept only for csv input.

Combining the UNOISE algorithm and the entropy correction

Sequences are stored as a data frame, with each row corresponding to a sequence record and the columns to the abundances (either total or per sample). The original Edgar’s (2016) function used by UNOISE to determine whether two sequences should be merged is:

βd=0.5α⋅d+1

where β(d) is the threshold abundance ratio of a less abundant sequence with respect to a more abundant one (from which it differs by distance d) below which they are merged. The distance d is the Levenshtein genetic distance measured in DnoisE with the Levenshtein module (https://maxbachmann.github.io/Levenshtein/) and α is the stringency parameter (the higher α, the lower the abundance skew required for merging two sequences).

The UNOISE algorithm sorts sequences by decreasing abundance and each one is compared with the less abundant ones. At each comparison, the distance between sequences (d) is computed and, if the abundance ratio between the less abundant and the more abundant sequence is lower than β(d), the former is assumed to be an error. In UNOISE terminology, the sequences form clusters, of which the correct one is the centroid and the remaining members are inferred to derive from the centroid template but contain errors. In his original paper, Edgar (2016) suggests constructing a table of centroids excluding low abundance reads, and then constructing a ZOTU table by mapping all reads (before the abundance filtering) to the centroids table using the same merging criterion but without creating new centroids. So, the original formulation of this algorithm gives priority to the abundance ratio over the genetic distance. The first, very abundant, sequences will “capture” rare sequences even if d is relatively high. Other, less abundant sequences may be closer (lower d) and still fulfill Edgar’s formula for merging the rare sequence, but this will never happen as the rare sequence will be joined with the very abundant one and will not be available for further comparisons. However, in the standard procedure of this algorithm implemented as UNOISE3 in the USEARCH pipeline (Edgar, 2010; https://drive5.com/usearch/), the reads are mapped to the centroid table using a similarity criterion (identity threshold in the otutab command), so in practice a distance criterion is used during the mapping.

DnoisE is a one pass algorithm, with no posterior mapping of reads to centroids (which is indeed repetitive, as reads have already been evaluated against the centroids when constructing the centroid table) and with a choice of merging criteria. If deemed necessary, low abundance reads can be eliminated previously or, alternatively, ESVs with one or a few reads can be discarded after denoising. Chimeric amplicons can likewise be eliminated before or after denoising. DnoisE follows previously used terminology (Turon et al., 2020; Antich et al., 2021) in which the correct sequences (centroids in UNOISE terms) are called “mother” sequences and the erroneous sequences derived from them are labelled “daughter” sequences. DnoisE provides different options for merging the sequences. Let PMS (potential “mother” sequence) and PDS (potential “daughter” sequence) denote the more abundant and the less abundant sequences that are being compared, respectively, and let d be the genetic distance between them. When the abundance ratio PDS/PMS is lower than β(d), the PDS is tagged as an error sequence but is not merged with the PMS. Instead, a round with all comparisons is performed and, for a given PDS, all PMS fulfilling the UNOISE criterion for merging are stored. After this round is completed, the merging is performed following one of three possible criteria: (1) Ratio criterion, joining a PDS to its more abundant PMS (lowest abundance ratio, corresponding to the original UNOISE formulation); (2) Distance criterion, joining a sequence to the closest (least d value) possible “mother”; and the (3) Ratio-Distance criterion, whereby a PDS is merged with the PMS for which the quotient β/β(d) (i.e., between the abundance ratio PDS/PMS and the maximal abundance ratio allowed for the observed d), is lowest, thus combining the two previous criteria. For each criterion, the best PMS and the corresponding values (ratio, d and ratio skew values) are stored. The user then has the choice to select one or another for merging sequences. As an option, if the user wants to apply only the Ratio criterion, each PDS is assigned to the first (i.e., the most abundant) PMS that fulfills the merging inequality and becomes unavailable for further comparisons, thus decreasing computing time. Figure 1 shows a conceptual scheme of this workflow process.

Figure 1 Scheme of the workflow of DnoisE.

Starting from an abundance-sorted sequence dataset, subsets of possible daughter sequences (PDS) and possible mother sequences (PMS) are selected as detailed in Fig. 2. For each subset, all PDS are compared with all compatible PMS (in terms of MDA and MMA). If the merging inequality is met, the values of the main parameters are stored. After all subsets have been evaluated, for each merging criterion the best PMS for each PDS is chosen and a sequence file is generated, together with a file with information on the merging process.

In addition, for coding markers such as COI, the codon position provides crucial additional information that must be taken into account. In nature, the third codon position is the most variable, followed by the first and the second position. This variation can be measured as entropy (Schmitt & Herzel, 1997) of the different positions. A change in third position is more likely to be a natural change (and not an error) than the same change in a second position, much less variable naturally. To our knowledge, no denoising algorithm incorporates this important information. We propose to use the entropy values of each codon position to correct the distance d in Edgar’s formula as follows:

dcorr=∑i=13di⋅entropyi.3/entropy1+entropy2+entropy3

where i is the codon position and d is the number of differences in each position. The dcorr value is then used instead of d in the formula. This correction results in a higher dcorr when a change occurs in a third position than in the first or second position, thus a sequence with changes in third positions will be less likely to be merged. In practice, as many changes occur naturally in third positions, this correction will lead to a higher number of ESVs retained that would otherwise be considered errors. Careful choice of entropy values is crucial, and it is recommended that they are adjusted for each marker and particular study. The values of entropy for each position can be obtained from the data (computed directly by the program) or added manually by the user.

Note that, when applying this correction, the Levenshtein distance is not used as it cannot consider codon positions. Instead, the number of differences is used. In practice, in aligned sequences with no indels both distances are equivalent. In addition, with the entropy correction, lengths should be equal when comparing two sequences. The dataset is thus analysed separately by sequence length sets. These sets must differ from the modal length (the modal sequence length can also be set using the -m parameter) of the complete dataset by n number of codons (groups of three nucleotides), as in general indels in coding sequences are additions or deletions of whole codons. A sequence differing from these accepted lengths is considered erroneous and removed. Sequences of the same length must be aligned for the algorithm to run properly.

Parallel processing

Parallel processing is a useful tool to increase speed when multicore computers are available. DnoisE implements parallel processing in the algorithm so the required time to run huge datasets decreases drastically as more cores are used. Parallel processing was applied using the multiprocessing module of Python3 (McKerns et al., 2011). A computational bottleneck of denoising procedures is their sequential nature, which is hardly parallelizable, and more so in the case of DnoisE that computes all comparisons before merging. In particular, a sequence that has been tagged as “daughter” (error) cannot be a “mother” of a less abundant sequence. Therefore, to compare a PDS to all its PMS requires that those more abundant sequences have been identified as correct before.

We incorporate two concepts, based on the highest skew ratio required for a sequence to be merged with a more abundant one. This is of course β(min(d)), where min(d) is one if entropy correction is not performed, and it equals the dcorr corresponding to a single change in the position with less entropy (position 2) if entropy is considered. From this maximal abundance ratio we can obtain, for a given potential “mother”, the maximal “daughter” abundance (MDA, any sequence more abundant than that cannot be a “daughter” of the former). Conversely, for a given “daughter” sequence we can obtain the minimum “mother” abundance (MMS, any sequence less abundant than that cannot be the “mother” of the former). The formulae are:

MDA=abundancePMS/βmind

MMA=βmind/abundancePDS

βmind=0.5α⋅1+1ORβmind=0.5α⋅ minentropyi⋅3/entropy1+entropy2+entropy3+1

The use of MDA and MMA simplifies the workload of the program as it greatly reduces the number of comparisons (a PMS will not be evaluated against sequences more abundant than the MDA, and a PDS will not be compared with sequences with less abundance than the MMA). Likewise, it allows for a parallel processing of sequences using the MDA as follows:

1- Sequences are ordered by decreasing abundance.

2- The first sequence is automatically tagged as a correct sequence.

3- MDA is calculated for this sequence (MDA_1).

4- All sequences with abundances between the first sequence and the MDA are, by definition, tagged also as correct sequences.

5- For the last sequence tagged as correct, the MDA is calculated (MDA_2).

6- Every sequence with abundance between the last correct sequence and MDA_2 is evaluated in parallel against all correct sequences that are more abundant than its MMA. Those for which no valid “mother” is found are tagged as correct, the rest are “daughter” (error) sequences.

7- Repeat steps 5 and 6 (i.e., calculating MDA_3 to n) until all sequences have been evaluated.

Figure 2 provides a conceptual scheme of this procedure. Note that, for each block of sequences that is evaluated in parallel, no comparisons need to be performed between them as they will never fulfill the merging inequality. After this process is completed, all sequences not labelled as “daughter” are kept as ESVs, and all “daughters” are merged to them according to the merging criterion chosen.

Figure 2 Schematic workflow of parallel processing of DnoisE.

When running in parallel, comparisons between sequences are computed in sets of sequences defined by their abundances. Using the Maximum Daughter Abundance (MDA) value, computed from the last correct sequence of the previous step, we can define sets of sequences that are compared in parallel with the previously tagged correct sequences.

DnoisE performance

A previous version of DnoisE was tested in Antich et al. (2021) on a COI metabarcoding dataset of marine benthic communities. The version used in Antich et al. (2021) implemented the same basic algorithm but was not curated for general use. For the present version, we have corrected bugs, made the program user-friendly, and added more settable options and features. The dataset consisted of 330,382 chimera-filtered COI sequences of 313 bp (all sequences had more than one read). They came from benthic marine communities in 12 locations of the Iberian Mediterranean coast (see (Antich et al., 2021) for details), and are available as a Mendeley Dataset (https://data.mendeley.com/datasets/84zypvmn2b/). DnoisE was used in Antich et al. (2021) in combination with the clustering algorithm SWARM, and was compared with the results of DADA2 denoising algorithm. Antich et al. (2021) also compared DnoisE with and without entropy correction, and obtained twice the number of ESVs with correction, while the proportion of erroneous sequences (defined as those having stop codons or substitutions in conserved positions) decreased to one half as compared with not correcting for codon position variation, as discussed in Antich et al. (2021).

Comparison with UNOISE3

We benchmarked the current version of DnoisE (with alpha = 5) against the current implementation of the UNOISE algorithm: UNOISE3 (USEARCH 32-bit, free version, with alpha = 5 and minsize = 2) on this same dataset. To be able to make a direct comparison, for UNOISE3 we didn’t perform an otutab step, rather, we recovered the ESVs and their abundance directly from the output files generated with -tabbedout and -ampout. As chimeric sequences were already removed from the dataset, and for the sake of comparability, we didn’t exclude the few sequences flagged as such by the chimera filtering procedure embedded in UNOISE3. The number of ESVs obtained was almost the same: 60,198 and 60,205, respectively, if no entropy correction was performed. In addition, 60,196 ESVs were shared (comprising > 99.999% of the total reads) among the two programs, confirming that DnoisE (without correction) and UNOISE3 were practically equivalent. For further analyses of the effect of entropy correction we will therefore compare DnoisE with and without this correction.

Running performance

We compared the run speed of DnoisE with and without entropy correction for the same dataset of sequences. We used different numbers of cores, from 1 to 59, for parallelization. We applied the entropy correction values from Antich et al. (2021).

Running DnoisE with just one core (without entropy correction) took about 29 h, decreasing sharply when using parallel processing with just a few cores. DnoisE took 4.5 h with 6 cores and 2.78 h with 10 cores. As a reference, the execution time of UNOISE3 (32-bit version, not parallelizable) without the otutab step was ca. 7 h, albeit this execution time is not directly comparable as UNOISE3 has a chimera filtering step embedded. Using entropy correction, run times increased (Fig. 3) as there is a higher number of comparisons needed because the MMA values are generally lower. This slows the process as any given PSD has more PMS to compare with. With entropy correction, DnoisE retrieved ca. twice the number of ESVs, further increasing run time. For the Ratio-Distance merging criterion, when entropy correction was performed, 16 cores were required for DnoisE to run at a similar time speed than 6 cores with no entropy correction (Fig. 3). Above 10 cores (without correction) or 20 cores (with correction), run times reached a plateau and did not further improve, while memory usage continued to increase steadily. A trade-off between both parameters should be sought depending on the cluster architecture and the dataset being run.

Figure 3 Time (blue) and memory (red) used by DnoisE to denoise and merge sequences with the Ratio-Distance criterion using different cores on a computer cluster.

Denoising using entropy correction (triangles and dashed line) is compared against no correction (circles and dashed line). Lines are computed using the geom_smooth() function of the ggplot2 package with method = ‘loess’.

Merging performance

Due to the practical impossibility of building a mock community of the complexity required with known COI haplotypes for multiple species, in order to compare the merging performance of the original formula of UNOISE with the entropy correction available in DnoisE, we performed a simulation following the procedure described in Turon et al. (2020), and using the same dataset of 1,000 “good” sequences from marine samples used in that study. The rationale was to start with a dataset of good sequences with realistic read abundance distribution, simulate sequencing errors at a given error rate (henceforth “error” amplicons), and then denoise the resulting dataset to recover the original one. In addition, in the present study we kept track of which original sequences produced each error amplicon and used this information to check if error sequences are merged or not with their “true” mother. We applied a random error rate per base of 0.005, which is intermediate among reported values for Illumina platforms (Pfeiffer et al., 2018; Schirmer et al., 2016). After the simulation, we removed all sequences with only one read. This resulted in a dataset with the 1,000 original sequences and 265,297 error sequences.

We used the DnoisE software with and without entropy correction (the latter equivalent to the UNOISE3 results, see above) to denoise the simulated dataset. The entropy values were automatically computed from the data by the program and we tested alpha values from 10 to 1 (from lowest to highest stringency level). The results showed a decreasing number of total remaining sequences with more stringent (lower) alpha values (Fig. 4). There was also a drop in the number of good sequences remaining as alpha diminished. Except for the less stringent alpha values, however, data denoised with entropy correction kept a higher number of true sequences. With entropy correction, they remained almost constant for alpha values of 5 or higher, and decreased at lower values. Without entropy correction, the number of true sequences started to decrease at alpha values below 8. On the other hand, the entropy correction procedure also retrieved a higher number of false positives (i.e., error sequences) at intermediate alpha values, but the vast majority of them could be removed by applying a minimum abundance filter of 10 reads (–min_abund 10).

Figure 4 Number of original (correct) sequences (red), total sequences (dark blue) and total sequences filtered by read abundance (light blue) retrieved by DnoisE with entropy correction (solid line) and without entropy correction (equivalent to UNOISE).

Values with abundance filtering were computed using a minimum abundance of 10 reads (–min_abund 10).

We also computed the match ratio, which is the ratio of sequences that merged with their “true” mothers divided by the number of merged sequences (Fig. 5). For alpha values of 6 or higher, the match ratio was close to 1 irrespective of the use of entropy correction or not, albeit it was slightly better without correction. At lower values of alpha, the match ratio decreased markedly for the Ratio merging criterion, and more so without correction, reaching values of ca. 75% at alpha =1. There were also marked differences in the three joining criteria (compared only for the runs with entropy correction). While the abundance Ratio criterion resulted in a strong decrease of the match ratio, using the Distance or the Ratio-Distance joining criteria, the match ratios remained close to 1 until values of alpha 3 and decreased slightly at alpha 2 and 1. Note that the different joining criteria do not affect the number of ESVs produced, but the number of sequences merged with each ESV and, thus, their relative abundances. By keeping track of which original sequence produced each error sequence, we could compare how the relative performance of the different methods changed with alpha values.

Figure 5 Match ratio (error sequences merged to their “true” mothers/total number of merged sequences) of DnoisE without entropy correction and abundance ratio joining criterion (equivalent to UNOISE) grey bars) and DnoisE with entropy correction.

For DnoisE with entropy correction the three merging criteria were compared, abundance ratio criterion (orange bars), the genetic distance criterion (blue bars) and the criterion based on the cocient between the abundance ratio and the β(d) (green bars).

While this simulated dataset may not be a perfect representative of true metabarcoding datasets, it nevertheless highlights the importance of choosing the correct parameters of both alpha and minimum abundance filtering values as well as the need of choosing the proper joining criterion, especially at more stringent denoising levels (lower values of alpha). Note also that the results can vary depending on the error rate (we acknowledge that applying an uniform error rate of 0.005 is a simplification). Alpha values of 5 have been proposed for datasets of this COI fragment (Elbrecht et al., 2018; Shum & Palumbi, 2021; Turon et al., 2020) using several lines of evidence, but none of these studies included entropy correction. In addition, a minimal abundance filtering step is deemed necessary (Elbrecht et al., 2018; Turon et al., 2020) but an adequate threshold should be determined in each case. With our dataset and the explored error rate, values of 4 for alpha and 10 for minimal abundance seem a good compromise between keeping ca. 95% good sequences and accepting only a few error sequences. Our results emphasize the importance of calibrating the parameters for each type of data using any available evidence, including mock community data when available. The flexibility of DnoisE can greatly facilitate this exercise in future studies.

Conclusions

DnoisE is a novel denoising program that can be incorporated into any metabarcoding pipeline. It is a stand-alone program that addresses exclusively the denoising step, so that users can apply their favourite programs at all other steps (e.g., chimera filtering, clustering…). Moreover, DnoisE is open-source code. Other programs used in metabarcoding pipelines also have open codes, such as DADA2 (Callahan et al., 2016), OBITOOLS (Boyer et al., 2016), SWARM (Mahé et al., 2015), or VSEARCH (Rognes et al., 2016). We strongly adhere to the open software concept for continuous and collaborative development of computing science and, in particular, in the metabarcoding field.

DnoisE is based on the UNOISE algorithm developed by Edgar (2016), but with three main improvements: first, it allows to select among different criteria for joining sequences to optimize the match ratio; second, it incorporates the option to perform an entropy correction for coding genes, thus keeping more true sequences with high natural variability in third nucleotide positions in the codon; third, it is parallelizable to take advantage of the cluster architecture of modern computers.

Our correction by entropy opens a new field of analysis of coding genes, considering the different natural variability between codon positions. The flexibility of DnoisE with its settable options make this program a good tool for optimizing parameters in metabarcoding pipelines and for running the denoising step at any desired point of the pipeline (before or after clustering sequences into MOTUs).

In the next few years, processors are expected to reach the minimum size permitted by quantum laws. Parallel processing is needed to optimize future computer performance (Gebali, 2011; Zomaya, 2005). DnoisE offers a new parallel processing algorithm based on the MDA (maximum “daughter” abundance) to run analyses in parallel by groups of sequences that do not need to be compared between them. Parallel processing allows users to run huge datasets in a fast way using multithread computers. In our example, when running with 10 cores, DnoisE took about 2.78 h to compute a large dataset. On the other hand, memory management can be critical when running a high number of cores and large datasets and should be considered when setting the running parameters. DnoisE is written in Python3, one of the most popular languages, so it is a good option for users who want to modify or customize the code. We indeed encourage new developments of this software.

We consider that DnoisE is a good option to denoise metabarcoding sequence datasets from all kinds of markers, but especially for coding genes, given the entropy differences of codon positions. More details, sample files and complete instructions are available at GitHub (https://github.com/adriantich/DnoisE).

Supplemental Information

Supplemental Information 1 Script for installing DnoisE

Click here for additional data file.

Supplemental Information 2 Examples of scripts for running DnoisE using different options

Click here for additional data file.

Supplemental Information 3 Scripts for comparing DnoisE and Unoise3

Click here for additional data file.

Additional Information and Declarations

Competing Interests

Author Contributions

Data Availability

Owen S. Wangensteen is an Academic Editor for PeerJ.

Adrià Antich conceived and designed the experiments, performed the experiments, analyzed the data, prepared figures and/or tables, authored or reviewed drafts of the paper, wrote the original code in Python, and approved the final draft.

Creu Palacín conceived and designed the experiments, prepared figures and/or tables, authored or reviewed drafts of the paper, and approved the final draft.

Xavier Turon and Owen S. Wangensteen conceived and designed the experiments, performed the experiments, analyzed the data, prepared figures and/or tables, authored or reviewed drafts of the paper, and approved the final draft.

The following information was supplied regarding data availability:

The open source code for DnoisE is publicly available at GitHub: https://github.com/adriantich/DnoisE.

The data set used to test the software is publicly available at Mendeley: Antich, Adrià; Palacin, Cruz; Wangensteen, Owen; Turon, Xavier (2021), “Dataset for ”To denoise or to cluster? That is not the question. Optimizing pipelines for COI metabarcoding and metaphylogeography””, Mendeley Data, V3, doi: 10.17632/84zypvmn2b.3.

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
