# Peer review of "DnoisE: distance denoising by entropy. An open-source parallelizable alternative for denoising sequence datasets"

_PeerJ, doi:10.7717/peerj.12758_

## Round 0.1 · original submission · Major Revisions

Dear Dr. Antich and colleagues:

Thanks for submitting your manuscript to PeerJ. I have now received two independent reviews of your work, and as you will see, the reviewers raised some concerns about the research. Despite this, this the reviewers are optimistic about your work and the potential impact it will have on research studying genome sequencing and assembly methodologies. Thus, I encourage you to revise your manuscript, accordingly, taking into account all of the concerns raised by both reviewers.

Please consider the relevance of your study in relation to other similar approaches. The novelties and superiority of DnoisE compared to these other approaches needs to bemade clearer.

The reviewers identified some problems with your code that need to be addressed.

Please consider adding automatic testing or address why you have not done so.

I agree the concerns of the reviewers, and thus feel that their suggestions should be adequately addressed before moving forward.

I look forward to seeing your revision, and thanks again for submitting your work to PeerJ.

Good luck with your revision,

Best,

-joe

·

Basic reporting

A software that uses the properties of coding genes to filter erroneous reads has been desired for some time now and the authors succeed in bringing this research forward.

The entropy corrected denoising algorithm, benchmark and discussion of DnoisE has already been published in Antich et al. 2021. Although they mentioned that it is a beta-version, it is not exactly clear what has been added to the software since then. The novelty seems to be the description of parallel processing, even though parallel processing was already available since the publication of Antich et al. 2021 in April. Maybe the authors should state more clearly what this manuscripts adds additionally.

Experimental design

The presented method is relevant, closes a current gap and is important for the further development of the field.

The entropy correction is a sound way to utilize the coding region properties of genes like cox1.

The available software on github does not provide any tests to show that the program actually works as intended. I suggest adding tests for core functionalities, since I have observed bugs during my investigation (see section 4).

Entropy correction has been benchmarked in Antich et al. 2021. The authors used a dataset which was filtered beforehand by excluding sequences that do not align properly due to, for example, indels or offsets. The authors mention on the lines 188-189 that in practice the distance is the same, if there are no indels and the sequences are aligned. This seems to be an important prerequisite for the usage of this software and should be stated more clearly. Depending on sample and primers, offsets might be frequent and many taxa do have indels in the cox1 fragment. It is also mentioned in Antich et al. 2021: “We used a dataset of fixed sequence length and eliminated misaligned sequences. The correction for codon position would be more complicated in the presence of indels and dubious alignments.“
-> Therefore I strongly recommend to mention in the manuscript and on the github README page, that such a prefiltering has to be done before using DnoisE.

Validity of the findings

The distinction between correct and wrong merged sequences seems generally reasonable. Although a representative mock community would be the only way to be somewhat sure that the results are valid. Such a mock community would need to have many specimens per species and a wide range of differing biomasses per specimen. I see that such a mock community is not available at the moment and might never be. Because of that I suggest adding a remark that this approach needs further investigation and other benchmarks.

Regarding the benchmark: How has it been decided which genetic code was used to translate the sequences from nucleotides to amino acids, if there were the same number of stop codons?

Has each denoised sequence been translated afterwards with the same genetic code across the different denoising strategies (DnoisE, Unoise3, dada2)? This is not clear from the benchmark of Antich et al 2021 and could lead to differing decisions regarding the correctness of a sequence.

Additional comments

Line 283: Change “Is it” to “It is”

Line 318-319: No samples provided

Some comments in the source code are not in english and should be translated.

I found some bugs as I tested DnoisE:
The -x option did not work.

python3 --version
Python 3.8.10

command: python3 ../DnoisE/src/DnoisE.py -i in.fa -o out.dout -F F -c 4 -e 0.3468,0.159,0.709 -y T -x 2

stdout:
starting to denoise
['-i', 'in.fa', '-o', 'out.dout', '-F', 'F', '-c', '4', '-e', '0.3468,0.159,0.709', '-y', 'T', '-x', '2']
Denoising in.fa file
Output files will be named out.dout*
Fasta output file: False
Running with 4 cores
E1: 0.3468
E2: 0.159
E3: 0.709
Is entropy taken into account: True
Traceback (most recent call last):
File "../DnoisE/src/DnoisE.py", line 31, in <module>
de.read_parameters(argument_list)
File "/home/user/reviews/peerj/DnoisE/DnoisE/src/denoise_functions.py", line 198, in read_parameters
self.initial_pos = int(current_argument)
ValueError: invalid literal for int() with base 10: '-x'

Therefore users are not able to use it on datasets that have a different then the default value. Since there is wide variety of primers and start positions, this seems to be crucial option to work as intended. Not every user will be able to do modify the default value in the source code, hence this needs to be fixed if the software should be used with a variety of datasets.

To_uniq_line_fasta.sh does not work. It also has a wrong usage message

entrpy.R does not work if the input file ends with .fas:
Error: unknown input file format!
Execution halted

At least on my machine, the entrpy.R script consumes quite quickly too much memory and the process gets killed, or I get the following error message (tested with a 500 MB text file):
Error in total + rebost : non-conformable arrays
Calls: entrpy
Execution halted

I would suggest fixing this issue and if not possible: mention the problem on the github README. This is of course dependent on the data and hardware, but if the user is warned beforehand he could shrink the data by only using sequences with size > 1, or try it on a computer with more RAM.



References:
Antich, A., Palacín, C., Wangensteen, O. S., & Turon, X. (2021). To denoise or to
cluster, that is not the question: optimizing pipelines for COI metabarcoding and
metaphylogeography. BMC Bioinformatics, 22(1), 177.
https://doi.org/10.1186/s12859-021-04115-6

·

Basic reporting

The authors present a software package, called "DnoisE", that implements a known denoising strategy (UNOISE2/3) in an open-source package, with the addition of a novel strategy that should increase the performance when analysing coding markers.
I consider this software a welcomed addition in the tool shed of the bioinformatician analysing metabarcoding data, in particular for its claim of being better suited for less popular markers, suck as COI.

The manuscript introduces the core concepts (Background), the algorithm (Workflow) and the benefits of parallel processing (Benchmark).

Experimental design

The "Workflow" paragraph can be expanded, as it should be noted that UNOISE is only available as a closed-source commercial package (USEARCH) and its description is only available as a non peer-reviewed pre-print, that focuses on UNOISE2 and - from the author's website - UNOISE3 is only described as an improved set of defaults (as far as I understood).

There is a lack of an analysis of the accuracy and sensitivity of the program if compared with its natural ancestor (UNOISE3) and its most used competitor (DADA2), and the "Benchmark".

The benchmark section describes briefly a comparison performed with a published dataset, but fails to provide the code used to run the packages and to evaluate the differences in stop codons.
To benchmark the novelty of the program (i.e. analysis of coding regions with information entropy) can be evaluated with a well known mock community.

Validity of the findings

The DnoisE program is an interesting software, but the manuscript does not provide a sufficient exploration of its application domain and its sensibility/sensitivity/F1 etc.

Additional comments

As a tool user and tool developer, I find the structure of the software and its repository quite poor and distant from the standards of packages released in the last decade.

As a user, it's legitimate to expect a facilitated installation, for example via Miniconda (I'm glad to provide guidance to package the software for Bioconda), and an author controlled set of versions ("releases" in github), to increase the reproducibility.

As a developer I found the code poorly structured and not documented, lacking input validation both for trivial usages (e.g. providing FASTQ files instead of FASTA files) and for more delicate formatting issues: the output of "usearch -fastx_uniques XYZ -sizeout" was not accepted if the sequence names contained non alphanumeric characters (e.g. ">MSED1-end.42;size=3370;").

The absence of automatic testing will increase the risk of introducing unexpected behaviours in future releases.

The software is the main output of this research, and I would welcome an improved adherence to widely adopted criteria for usability and reproducibility.


VSEARCH, published in this journal on 2016, is a great example: https://github.com/torognes/vsearch.

---

## Round 0.2 · Major Revisions

Dear Dr. Antich and colleagues:

Thanks for revising your manuscript. The reviewers are mostly satisfied with your revision (as am I). Great! However, there are some new issues that arose as well as some previous criticisms still left to entertain. Please address these ASAP so we may move forward with your work.

The user feedback by reviewer 1 is invaluable...please address their concerns and improve the program’s implementation and usability.

Best,

-joe

·

Basic reporting

-

Experimental design

-

Validity of the findings

-

Additional comments

At first, I was very happy to see the improvements of DnoisE and I think the authors did in general a great job addressing our concerns.

However, I have noticed a few things that I would like to describe:

I tried DnoisE with the provided example files and also with my own data. Using my own data, there was no difference between the output if entropy correction was enabled (-y) and if not. It resulted in the same sequences. This seems to me as a strange result and I wonder how this could happen? I would offer to provide my test data. If needed, please contact me via email: n.noll@leibniz-zfmk.de
The example from your github repository did however show different results. But not as expected. Here we find less sequences if entropy correction was enabled. This is contrary to the findings of Antich et. al 2021 and I wonder if this is due to the different dataset or is it because of a change in the entropy calculation?


After I installed the program with ./install.sh the binary did not work with the following error message:
./DnoisE -h
Traceback (most recent call last):
File "src/DnoisE.py", line 12, in <module>
ModuleNotFoundError: No module named 'denoise_functions'
[15164] Failed to execute script 'DnoisE' due to unhandled exception!

It does however work if I use python3 ./src/DnoisE.py. Therefore, I suggest fixing the issue with the binary.


The result output *denoising_info.csv seems as if it is not correctly generated. Another problem is that the output files are neither explained in the current manuscript and also not on the github page. I would expect a brief explanation of what every column name means. For example xavier_criteria.
The second column “mother_d” seems to be the name of the mother sequence, whereas d shows otherwise an integer.
The column “mother_ratio” holds the name of the mother sequence and is not a ratio


The help message displays:
y --entropy_correction a distance correction based on entropy is performed (see ENTROPY CORRECTION below). If set to F, no correction for entropy is performed (corresponding to the standard Unoise formulation)

There is no “below” and the entropy correction is not explained further. It is also misleading that no correction is performed if -y is set to F. The default of DnoisE is to not use entropy correction. Using the -y option enables it. Since this seems to be just a flag argument, it cannot be set to false and it would also be unnecessary.


If the input file is of type fasta, a semicolon is needed at the end of the header. Since the output files from usearch or vsearch do not include semicolons at the end, I would recommend allowing users to use files without these.


I could still find no automated tests that ensure the correctness of core functions or output results.


L106-L111: It should be added that aligned sequences are needed. The entropy correction is the main novel feature of DnoisE and using it without aligned sequences this feature does not work properly.

·

Basic reporting

I would like to thank the authors for the improvements and for kindly addressing my questions.

I recommend a more detailed explanation of the core algorithm (as schematized in Figure 1), and a clearer comparison (in a dedicated paragraph) with the procedure previously published in Antich 2021 and UNOISE itself.

I think added clarity in this aspect is pivotal for a better understanding of the paper, especially for those who already implemented the methodology as originally described.
In particular, I recommend not to refer to the software used in Antich 2021 as "beta", as it was in fact used in production for the paper itself, and according to the authors, the changes are not expected to produce different findings.

Experimental design

no comment

Validity of the findings

In the "DnoisE performance" paragraph I suggest a more explicit comparison with Unoise.
A Venn diagram with the shared identical sequences will be a good addition.

The mention of future work with mock communities is sound and appropriate, yet the increased number in ESV detected with entropy correction can include false positives.
Considering that the method was applied previously, and the entropy correction is the unique feature of the presented tool, I think that a formal validation of the entropy-correction algorithm is required, at least with simulated reads comparing the effect of the correction and of naive approaches.

---

## Round 0.3 · accepted · Accept

Dear Dr. Antich and colleagues:

Thanks for revising your manuscript based on the concerns raised by the reviewers. I now believe that your manuscript is suitable for publication. Congratulations! I look forward to seeing this work in print, and I anticipate it being an important resource for groups studying genome sequencing and assembly methodologies. Thanks again for choosing PeerJ to publish such important work.

Best,

-joe

---

## Author Rebuttal · Round 0.3

Dear editor,

We are very grateful for the invaluable comments and suggestions by the reviewers, which have significantly contributed to enhance our software and our manuscript. We have now produced an improved version of the DnoisE software, which not only fixed all bugs and issues pointed out by the reviewers, but is also equipped with more customizable options, making DnoisE a more versatile tool. We are submitting a new version of our article, including a detailed analysis of a simulated dataset of error-affected sequences, which we think will help to clarify the comparison between the results of our new software and Unoise. Our answers to all reviewer's comments are detailed below. We hope that this version meets the standards to be published in PeerJ. Thank you very much for your consideration,

Owen S. Wangensteen (in behalf of all authors)

Comments from the reviewers

Reviewer: Andrea Telatin

**Basic reporting**
I would like to thank the authors for the improvements and for kindly addressing my questions.
I recommend a more detailed explanation of the core algorithm (as schematized in Figure 1), and a clearer comparison (in a dedicated paragraph) with the procedure previously published in Antich 2021 and UNOISE itself.
I think added clarity in this aspect is pivotal for a better understanding of the paper, especially for those who already implemented the methodology as originally described.
In particular, I recommend not to refer to the software used in Antich 2021 as "beta", as it was in fact used in production for the paper itself, and according to the authors, the changes are not expected to produce different findings.

The version used in Antich et al (2021) was indeed based on the same algorithms and the results are not expected to differ. However, that was a much less mature version, with few parameterization options and not user friendly. This is now explained in the text (first paragraph of "DnoisE performance section"). In addition, there were several bugs that have been corrected and many new features (f.i., minimal read abundance filtering, sequence length filtering) and new sources (f.i., the Levenshtein distance module) have been added. Antich et al. (2021) only used the software and did not compare it with Unoise in a quantitative way, which is now presented in the manuscript as described below. There was also no explanation of the program workflow and processing (Figs 1 and 2 of the present manuscript). So we feel that, albeit the core functions were basically the same, a paper was required to present the software properly and make it available to a wider audience.

**Validity of the findings**
In the "DnoisE performance" paragraph I suggest a more explicit comparison with Unoise. A Venn diagram with the shared identical sequences will be a good addition.

The mention of future work with mock communities is sound and appropriate, yet the increased number in ESV detected with entropy correction can include false positives. Considering that the method was applied previously, and the entropy correction is the unique feature of the presented tool, I think that a formal validation of the entropy-correction algorithm is required, at least with simulated reads comparing the effect of the correction and of naive approaches.

We thank the reviewer for this suggestion. It is true that a mock community of true haplotypes would be a good option to validate DnoisE but building such community with the required complexity would be a cumbersome endeavour. Following this comment we have now added a comparative analysis with a simulated dataset of "good sequences" affected by random errors, following Turon et al. (2020), including tracking from which "true" sequence each simulated error amplicon derived. We have changed the structure of the "DnoisE performance" section to show the results of this comparison, adding a "merging performance" heading, and incorporated two new figures to the ms.

Note also that we are now referring to the algorithm presented by Edgar (2016) as the UNOISE algorithm, to distinguish it from the software that applies it (UNOISE3 being the current version).

As UNOISE3 is not open source, has a chimera filtering procedure embedded, and is based on a posterior assignment of reads using -otutab, we could not directly compare the two programs. Instead, we first compared our DnoisE without entropy correction (which uses the same formula of Edgar 2016) with UNOISE3 by picking the information on sequence merging from the -ampout and -tabbedout files generated by UNOISE3, and without an -otutab step. We could verify that both programs produced the same results (99.99% of ESVs coincident involving 99.999% or reads). Once validated, we went on using DnoisE with and without entropy correction, to evaluate the entropy-correction algorithm at different alpha values, as our software is more flexible and allowed us to program features such as the tracking of original sequences.

Using this simulation we could compare not only between the results retrieved using or not entropy correction in terms of true and false positives but also between the different merging criteria that DnoisE can perform. This simulation showed clearly the potential of DnoisE over UNOISE3 and, particularly, the usefulness of its flexibility in choosing the adequate denoising strategy for a given dataset. This exercise also led us to add the --min_abund parameter to DnoisE which we think is a useful option not seen in other software.

The simulation dataset consists of 1,000 sequences considered error-free and with realistic abundance distribution, referred to as 'original' sequences, obtained from marine communities in Turon et al (2020). With this dataset we simulated sequencing errors, at

0.005 error rate (datum taken from the literature as representative), and obtained 265,297 error sequences. For the generated sequences ('error sequences' hereinafter) we kept in their ids the information of which was the original sequence from which they derived. With this information we could calculate the amount of error sequences that are denoised and if they were merged with their "true" original sequence or not.

As shown in results, entropy correction allows to retain more 'good' sequences but also some 'error' amplicons. We also suggest a filter of minimum abundance of 10 reads which eliminates most of the retained error sequences. To perform this final filter we have enabled an abundance filtering option (-r --min_abund; set as zero as default) to specify the minimum amount of reads required to retain a sequence.

Reviewer: Niklas Noll

At first, I was very happy to see the improvements of DnoisE and I think the authors did in general a great job addressing our concerns.
However, I have noticed a few things that I would like to describe:
I tried DnoisE with the provided example files and also with my own data. Using my own data, there was no difference between the output if entropy correction was enabled (-y) and if not. It resulted in the same sequences. This seems to me as a strange result and I wonder how this could happen? I would offer to provide my test data. If needed, please contact me via email: n.noll@leibniz-zfmk.de
The example from your github repository did however show different results. But not as expected. Here we find less sequences if entropy correction was enabled. This is contrary to the findings of Antich et. al 2021 and I wonder if this is due to the different dataset or is it because of a change in the entropy calculation?

We sincerely thank the reviewer for his thorough job testing our software. During the restructuring of the software for the last revision, entropy correction was inadvertently disabled (!). This has been fixed now. We have also detected other bugs that have been solved.

An important change that we performed in the last version was a filtering of sequence length when using entropy correction. This filtering step is necessary because entropy correction can only be performed to aligned sequences of the same length. Without entropy correction this is not necessary as the Levenshtein distance can be calculated even if there are indels. We also wanted to accommodate the natural variability in sequence length of some markers. This variability, for coding genes, is usually due to indels of one or several codons, and thus length differences are multiple of three nucleotides. With our COI fragment, in our experience with several datasets, ca. 95% of all sequences obtained have the modal length of 313 (modal length is customizable), but there is some variability among different species. DnoisE with entropy correction thus accepts sequences of (modal length) $\pm$ 3*n, being n the number of codons of difference. All other sequence lengths are considered erroneous and the corresponding reads are eliminated. This procedure is now explained in the ms (in the paragraph before "Parallel processing").

The length filter explains the effect detected by the reviewer with the DnoisE-test dataset. This dataset included sequences of different lengths. There were 14,073 sequences of which 13,291 had 313 bp, and 148 more had "compatible" lengths (313 ± 3*n). On the other hand, 1,233 sequences had unaccepted lengths and were removed when entropy correction was performed. This results in 1,703 ESVs without entropy correction and 1,785 with it. The increase in ESVs obtained as a result of entropy correction is almost compensated by the fact that, without entropy correction, 1,233 more sequences of unacceptable lengths are included. For large datasets the effect is negligible and more ESVs are retained with entropy correction. Note that in Antich et al 2021 we used a test dataset of sequences of uniform length (313 bp) and therefore the length filter was not necessary. It has only been implemented in the present version. We now explain in the README.md file of GitHub the potential effect of excluding sequences, and we have also added another test dataset including only the 13,291 sequences with 313 bp. So, the interested user can compare the results.

After I installed the program with ./install.sh the binary did not work with the following error message:
./DnoisE -h
Traceback (most recent call last):
File "src/DnoisE.py", line 12, in <module>
ModuleNotFoundError: No module named 'denoise_functions'
[15164] Failed to execute script 'DnoisE' due to unhandled exception!
It does however work if I use python3 ./src/DnoisE.py. Therefore, I suggest fixing the issue with the binary.

In the reviewed version, the binary was supposed to be produced by pyinstaller, however we have now changed it to nuitka (https://nuitka.net/). It takes more time but produces less errors. We have now also changed the module used for the computation of the Levenshtein distance,which was the origin of many problems in different systems. We now use the module from maxbachmann (https://github.com/maxbachmann/Levenshtein) instead of the one from ztane (https://github.com/ztane/python-Levenshtein). The binary produced is ./scr/DnoisE.bin and the installer should now work in most systems.

The result output *denoising_info.csv seems as if it is not correctly generated. Another problem is that the output files are neither explained in the current manuscript and also not on the github page. I would expect a brief explanation of what every column name means. For example xavier_criteria.
The second column "mother_d" seems to be the name of the mother sequence, whereas d shows otherwise an integer.
The column "mother_ratio" holds the name of the mother sequence and is not a ratio

The reviewer is right, we have now updated the info.csv column names and we describe this output file in detail in the Github README.md

The info.csv returns information on how the sequences have been merged. 'Mother' sequences have only its id name with no other information. For each 'daughter' sequence the following information is given.

'daughter' -> the sequence from which the information is retrieved

'mother_d -> the mother sequence corresponding to the 'd' criteria

'd' -> the d value corresponding to the comparison between daughter and mother_d

'mother_ratio -> the mother sequence corresponding to the 'ratio' criterion

'ratio' -> the ratio value corresponding to the comparison between daughter and mother_ratio

'mother_ratio_d -> the mother sequence corresponding to the 'ratio_d' criterion

'ratio_d' -> the ratio_d value corresponding to the comparison between daughter and mother_ratio_d

If a entropy correction is performed, the following information is also given:

'difpos1' -> number of differences in position 1 of the codon corresponding to the mother_ratio_d comparison

'difpos2' -> number of differences in position 2 of the codon corresponding to the mother_ratio_d comparison

'difpos3' -> number of differences in position 3 of the codon corresponding to the mother_ratio_d comparison

'dtotal' -> number of total differences corresponding to the mother_ratio_d comparison

'betacorr' -> beta value corrected by entropy values corresponding to the mother_d comparison.

The latter value is used during the process to choose the mother_d sequence and we think that it could be useful if future changes on the correction formula are performed by users.

The help message displays:
y --entropy_correction a distance correction based on entropy is performed (see ENTROPY CORRECTION below). If set to F, no correction for entropy is performed (corresponding to the standard Unoise formulation)
There is no "below" and the entropy correction is not explained further. It is also misleading that no correction is performed if -y is set to F. The default of DnoisE is to not use entropy correction. Using the -y option enables it. Since this seems to be just a flag argument, it cannot be set to false and it would also be unnecessary.

There was a mistake in the  --help message. It is true that it is just a flag argument. It is now solved.

If the input file is of type fasta, a semicolon is needed at the end of the header. Since the

output files from usearch or vsearch do not include semicolons at the end, I would recommend allowing users to use files without these.

Solved

We have now created the ./src/test_DnoisE.py script that calls .json files to run a small dataset for the core functions and to compare with expected results. It can be run from the terminal as follows:

python3 -m unittest test_DnoisE.py

L106-L111: It should be added that aligned sequences are needed. The entropy correction is the main novel feature of DnoisE and using it without aligned sequences this feature does not work properly.

As detailed above, DnoisE is now able to work with sequences of different lengths, and it will automatically remove sequences of unacceptable lengths. Details about the sequence length requirements are now better explained in the ms.

Note that, following a request from the first reviewer, the ms includes now a simulation study to compare the performance of the program with and without entropy correction (the latter equivalent to the UNOISE3 software) in terms of good sequences recovered, false positives, and correctness of the merging criteria. We have changed the structure of the "DnoisE performance" section accordingly, adding a "Merging performance" heading, and incorporated two new figures to the ms.

Note also that we are now referring to the algorithm presented by Edgar (2016) as the UNOISE algorithm, to distinguish it from the software that applies it (UNOISE3 being the current version).